# Liquid–Liquid Phase Separation of Biomacromolecules and Its Roles in Metabolic Diseases

**DOI:** 10.3390/cells11193023

**Published:** 2022-09-27

**Authors:** Zhihao Chen, Ying Huai, Wenjing Mao, Xuehao Wang, Kang Ru, Airong Qian, Hong Yang

**Affiliations:** 1Department of Obstetrics and Gynecology, Xijing Hospital, The Fourth Military Medical University, Xi’an 710032, China; 2Lab for Bone Metabolism, Xi’an Key Laboratory of Special Medicine and Health Engineering, School of Life Sciences, Northwestern Polytechnical University, Xi’an 710072, China; 3Key Lab for Space Biosciences and Biotechnology, Research Center for Special Medicine and Health Systems Engineering, School of Life Sciences, Northwestern Polytechnical University, Xi’an 710072, China; 4NPU-UAB Joint Laboratory for Bone Metabolism, School of Life Sciences, Northwestern Polytechnical University, Xi’an 710072, China

**Keywords:** Alzheimer’s disease (AD), liquid–liquid phase separation (LLPS), membraneless organelles (MLOs), metabolic bone diseases (MBDs), type 2 diabetes mellitus (T2DM)

## Abstract

Liquid–liquid phase separation (LLPS) compartmentalizes and concentrates biomacromolecules into liquid-like condensates, which underlies membraneless organelles (MLOs) formation in eukaryotic cells. With increasing evidence of the LLPS concept and methods, this phenomenon as a novel principle accounts for explaining the precise spatial and temporal regulation of cellular functions. Moreover, the phenomenon that LLPS tends to concentrate proteins is often accompanied by several abnormal signals for human diseases. It is reported that multiple metabolic diseases are strongly associated with the deposition of insoluble proteinaceous aggregating termed amyloids. At present, recent studies have observed the roles of LLPS in several metabolic diseases, including type 2 diabetes mellitus (T2DM), Alzheimer’s disease (AD), and metabolic bone diseases (MBDs). This review aims to expound on the current concept and methods of LLPS and summarize its vital roles in T2DM, AD, and MBDs, uncover novel mechanisms of these metabolic diseases, and thus provide powerful potential therapeutic strategies and targets for ameliorating these metabolic diseases.

## 1. Introduction

Liquid–liquid phase separation (LLPS) is a reversible and dynamical biophysical process where homogeneous biomacromolecules spontaneously de-mix into two coexisting liquid phases (a condensed phase and a dilute phase) through transient multivalent macromolecular interactions [1,2]. Currently, LLPS is reported to be considered the underlying of multiple biological processes, especially for the formation of membraneless organelles (MLOs), such as processing bodies (P-bodies), stress granules, and nucleolar. In fact, LLPS tends to compartmentalize and concentrate biomacromolecules into liquid-like condensates, which underlies MLO formation to explain the self-assembly process of subcellular structures [3]. LLPS also serves as an important natural defense mechanism in response to external various stimuli in living cells [4]. It is reported that LLPS is associated with the pathogenesis of multiple human diseases, such as neurodegeneration, infectious diseases, cancer, and aging diseases [5,6]. These recent reports have constructed a new framework to describe various biological processes and reexamine biological phenomena from the view of LLPS.

At present, the prevalence of metabolic diseases is still increasing globally, which represents a heavy public health burden. However, the in-depth mechanism of metabolic diseases is not yet completely understood. Metabolic diseases usually disrupt the critical biochemical reactions of cells, including the processing or transport of proteins (amino acids), carbohydrates (sugars and starches), or lipids (fatty acids) [7]. Most importantly, several metabolic diseases are strongly associated with amyloid depositions which are insoluble proteinaceous aggregates depositions. Interestingly, emerging evidence suggests that LLPS is one of the crucial ways in which protein aggregates into amyloid deposits [8]. Therefore, LLPS may emerge as a new mechanism underlying these metabolic diseases. In this review, we expound on the current concept and methods of LLPS and summarize the various ways in which they impact cellular metabolism processes and metabolic diseases. We present emerging evidence that LLPS is associated with metabolic diseases including type 2 diabetes mellitus (T2DM), Alzheimer’s disease (AD), and metabolic bone diseases (MBDs), which provides new insights into the mechanism and potential therapeutic strategies and targets for ameliorating these metabolic diseases.

## 2. Major Milestones of Liquid–Liquid Phase Separation Development

Although increasing research on LLPS has identified its essential roles in physiology and diseases, it also experienced a tortuous development history. The following brief introduction will discuss the research development and milestone achievements of liquid–liquid phase separation (Figure 1).

As early as 1899, Wilson first proposed the existence of droplets in cells [9]. He found that different fluids inside the cells could form suspended emulsion droplets according to the observation of sea urchin eggs. Although his view was largely unrecognized at the time, it also foreshadows the arrival of the exploration of LLPS in the biological field. Until 2009, Brangwynne observed liquid-like P granules in *Caenorhabditis elegans* embryo cells and first noticed the roles of LLPS in biology, which also explains how cells independently and orderly conduct biochemical reactions [10]. In 2012, Rosen’s team found that some multivalent proteins tended to undergo LLPS and assembled as liquid-like droplets during cytoskeleton formation, which suggested that low-complexity regions were key structures prone to LLPS [11]. Meanwhile, some studies identified multiple MLOs formed by biomacromolecule LLPS in various cells, such as nucleoli, Cajal bodies, and PML nuclear bodies in the nucleus [12], as well as SGs in the cytoplasm [13]. LLPS is an important organizing principle and theoretical basis of MLOs, which explains the regulation mechanisms of MLOs in their assembly, composition, and function [14]. In 2016, it is reported that the localization of ribonucleoprotein (RNP) particles involved in LLPS was achieved through the regulation of the RNA competition mechanism. In 2018, Zhang et al. found that *Mtorc1* regulated the autophagic degradation of PGL particles through LLPS, thereby affecting the developmental process of *C. elegans* [15]. In the same year, Shan et al. revealed the molecular mechanism that LLPS regulated the positioning of cell fate determinants [16]. In 2019, Gibson et al. demonstrated that the intrinsic LLPS capacity of the chromatin polymer played a major role in the organization and regulation of eukaryotic genomes [17]. Recently, Elaine and colleagues have reported that filaggrin-driven LLPS participated in the formation of the human skin barrier by regulating the epidermal structure and function [18]. These studies illustrated the various regulatory roles of LLPS in cellular biological processes. Despite the essential physiological roles of LLPS, the misassembly of RNA/proteins driving aberrant processes of LLPS results in the pathological process of multiple diseases, such as cancer, autoimmune diseases, and neurodegenerative diseases [4]. Recently, the outstanding achievements of LLPS and its potential applications in drug development have received unprecedented attention. Some proponents even said that LLPS might rewrite the rules of drug development [19].

## 3. Representative Research Methods of Liquid–Liquid Phase Separation

At present, LLPS has become a research hotspot in the field of biology. Therefore, the research methods of LLPS are also gradually diversified. Here, we briefly introduce the existing representative strategies of LLPS based on the in vitro and in vivo reported research, respectively.

### 3.1. In Vitro

It is easy to observe the process of LLPS and control the concentration and environmental conditions of each component in vitro. Thus, diverse microscopies are increasingly applied to determine the characteristics of liquid-like droplets formed by LLPS. For example, differential interference contrast (DIC) imaging is a representative method to visualize the properties of droplets, which can present the coexistence of two or more phases. Besides, fluorescence correlation spectroscopy (FCS) is considered another ideal tool to estimate the diffusion capacity of a single molecule inside the LLPS droplet [20]. FCS is always used to detect sparsely labeled and highly mobile components as well as droplet dilution. Atomic force microscopy (AFM) can describe the properties of biological condensate materials, such as viscosity, pore size, elasticity, and other parameters. Zeng et al. measured the mechanical properties of postsynaptic density (PSD) droplets to monitor individual phase performance by AFM [21]. Furthermore, liquid-phase transmission electron microscopy (LP-TEM) can enable direct visualization and real-time observation of liquid-like droplets formation to discover and renew biological assembly mechanisms [22]. Moreover, the fluorescent labeling and the dynamic imaging of liquid-like droplets are powerful methods to study the mechanisms of LLPS formation. In addition, the turbidity measurement assay is also a popular intuitive detection method for LLPS in vitro [23,24]. The components in the solution can scatter visible light from tens to hundreds of nanometers in diameter, which could be measured by optical density. Notably, this method just only detects the components in a droplet, the observation of the droplet shape, size and formation principle still requires a combination with microscopy [25]. Furthermore, centrifugal precipitation is also another common detection strategy of LLPS. We can observe transparent droplets that differ from the sediment and assess the proteins in different phases through centrifugation precipitation [26]. The light phase and the dense phase were separated by centrifugation, and then their concentration is measured by spectroscopy. Fluorescence recovery after photobleaching (FRAP) allows to capture of the exchange of substances between dense and dilute phases and observe the constant dynamic change process of LLPS, which is increasingly used to demonstrate molecular motion inside droplets [10]. Additionally, the optoDroplet is a tool that uses light to manipulate matter inside living cells and has begun to explain how proteins assemble into different liquid and gel-like solid states, a key to understanding many critical cellular operations. The optoDroplet tool is starting to allow us to dissect the rules of physics and chemistry that govern the self-assembly of MLOs. Importantly, we have only introduced the most common research methods of LLPS and MLOs. There are many more approaches to examine LLPS and MLOs, such as passive microrheology, active microrheology, cryoelectron tomography, nuclear magnetic resonance, capillary flow experiments, microfluidic tools as well as Corelets and PixELL platforms [27]. The existing research methods of LLPS in vitro are diversified (Figure 2), and more accurate detection technologies still need to be developed in the future.

### 3.2. In Vivo

The research methods of LLPS in vivo are more complicated compared with that in vitro. The high protein concentration is one of the important prerequisites for LLPS in cells. Therefore, overexpression of LLPS-triggering proteins is the common manner to drive and detect LLPS in vivo. At present, it is believed that the accepted criteria for a phase separation structure are the formation of a spherical structure, the ability to fuse, and the ability to recover from photobleaching [25]. The recovery time of FRAP not only depends on the protein/RNA concentration but also on the droplet size and the bleaching area [25]. Therefore, combining it with the other methods is more accurate for LLPS detection (Figure 2).

To identify the property of LLPS in vivo, Delarue et al. developed a homomultimeric scaffold fused with a fluorescent protein, named genetically encoded nanoparticles (GEMS), and used as an effective probe in the cytoplasmic matrix. The probe can evaluate the condensate porosity and parameters in the cellular environment [28]. Compared with the numerous research strategies of LLPS in vitro, how determining the physical and chemical properties of phase separation droplets in vivo still needs further exploration. In addition, it is also needful to explore several novel methods to explore the biological functions of LLPS in cells.

## 4. Liquid–Liquid Phase Separation Underlies MLOs Formation

It is well recognized that LLPS of biomacromolecules have emerged as a biophysical basis for the formation of MLOs in living cells [10]. Ubiquitously, MLOs in eukaryotic cells modulate a variety of physiological and pathological traits through multiple ways, which are closely related to the physical properties, types, and intracellular localization of MLOs [29]. Moreover, MLOs formed by LLPS are broadly distributed in the cytoplasm, nucleus, and membrane [30,31]. In this section, we mainly review the biological function of the MLOs localized in the cytoplasm such as stress granules (SGs), processing bodies (P-bodies), as well as in the nucleus including nucleoli, paraspeckles, PML bodies, and Cajal bodies. Table 1 displayed representative MLOs with different cellular localization and their function.

### 4.1. Cytoplasmic-Localized MLOs

Cytoplasmic-localized MLOs are dynamically assembled by the LLPS driving the temporarily untranslated RNAs and proteins, which coalesce into a concentrated state (the condensed phase) in the cytoplasm. Prominent examples of cytoplasmic-localized MLOs mainly include the stress granules (SGs), the processing bodies (P-bodies), the RNA transport granules, and the germ granules.

The stress granules (SGs) are a predominant type of cytoplasmic-localized MLOs formed by the crowded protein and RNA. The SGs immediately start to accumulate and regulate the mRNA utilization in eukaryotic cells under stress, which is essential for maintaining cell integrity and intracellular homeostasis [34,50]. Additionally, SG components mainly include aggregation-prone RNA binding proteins (RBPs), protein kinases, RNA helicases, structural constituents of ribosomes, calcium-binding proteins, hydrolases, and cytoskeletal proteins [51,52]. Moreover, dynein, microtubules [53], and various nucleocytoplasmic shuttling RBPs (TIA-1, TIAR, and HUR) [51] assist in the SGs secondary aggregation and assembly, which determines the speed and size of SG assembly. Moreover, the SGs are highly dynamic in nature, assembling, and dissembling quickly upon stress induction or stress disappearance, respectively. Their dynamic properties are mainly highlighted by the cytoskeleton system, which is a scaffold for SGs’ dynamic maintenance and movement [54]. Numerous researchers found that maintaining a proper SG dynamic might be a potential strategy to ensure cellular homeostasis and normal biological function in the living cell [55]. Thus, the normal dynamics of SGs play an important role in responding to stress stimuli, which can otherwise induce various human diseases.

The processing bodies (P-bodies) are the highly conserved cytoplasmic foci with properties of liquid droplets, which are formed by LLPS and are primarily composed of translation-arrested RNAs and RBPs related to mRNA decay [37,56]. Moreover, the P-bodies purification revealed that multiple RBPs including HNRNPU, IGF2BP1, DHX9, and HNRNPQ were the core components of P-bodies [57,58]. Additionally, P-bodies are distinct from SGs in multiple aspects, including the formation conditions, morphology, function, as well as components. Specifically, unlike SGs being exclusively stress-induced, the P-bodies are constitutive in some cells and nevertheless increase in size and number in response to stress [59]. Therefore, we conclude from these studies that the P-bodies exert multiple regulatory roles in the post-transcriptional processes, translation repression, and mRNA decay machinery.

In addition to SGs and P-bodies, there are still other well-studied cytoplasmic-localized MLOs, including germ granules, RNA transport granules, P granules, and the Balbiani body. For example, the germ granules and the P granules are conserved condensates enriched for RNA and RBPs in the germ cell cytoplasm, which play essential roles in the mRNA translation during gametogenesis and embryonic development [60]. The Balbiani body, also called a membraneless ball of mitochondria, contains various biomacromolecules and numerous membranous organelles. The Balbiani body is widely present in the majority of mammal oocytes [61]. In summary, these findings emphasize the important roles of LLPS in the formation and maintenance of cytoplasmic-localized MLOs and confirm the biological function of MLOs in cell growth and development.

### 4.2. Nuclear-Localized MLOs

In addition to the multiple cytoplasmic-localized MLOs, LLPS also is important for driving the assembly of various nuclear-localized MLOs such as nucleoli, Cajal bodies, and nuclear speckles, and underlies their biogenesis. The condensates within the nucleus could directly interact with chromatin, and thus potentially control its organization and gene expression. Moreover, the biomacromolecules and their multiple unique domains help to build these nuclear-localized condensates in the nucleus. In the following section, we will detail the assembly of the proteins/RNA in multiple nuclear-localized condensates and discuss the biological functions of the nuclear-localized MLOs.

The nucleolus is the most prototypical and prominent nuclear MLO. Nucleolus forms around the chromosome regions containing stretches of tandem ribosomal DNA (rDNA) gene repeats, known as nucleolar organizer regions (NORs) [62]. There is evidence that the nucleolus is formed through LLPS by its macromolecular components and exerts dynamic and liquid-like physical properties which might facilitate functions of the nucleolus in ribosome biogenesis and cellular stress sense [63]. Indeed, the nucleolus is composed of various RNA and hundreds of different proteins including RBPs. Nucleolin, a multifunctional stress-responsive RBP, is abundant in the nucleolus. It is reported that nucleolin could participate in rDNA transcription, rRNA maturation, ribosome assembly, and nucleocytoplasmic transport. Furthermore, nucleolin contains four RNA binding motifs, which indicates that nucleolin could undergo LLPS through its multivalent interactions with many other RNAs, and thus mediate the assembly of the nucleolus [64]. Besides, another nuclear protein nucleophosmin (NPM) has been confirmed to be able to facilitate the LLPS process in the multilayered structure of the nucleolus [65]. Furthermore, NPM harbors a low-complexity domain bound by poly (GR) and poly (PR), which could alter the LLPS properties of NPM and thus influence nucleolar dynamics in cells [66]. In brief, the LLPS triggered by several nuclear proteins plays crucial roles in the formation and biological function of the nucleolus.

Nuclear speckles, another well-studied MLO formed by LLPS in nuclear, exhibit dynamic and irregular shapes. Nuclear speckles are subnuclear structures enriched in the RBPs involved in splicing, which are located in the interchromatin regions of the nucleoplasm in mammalian cells [67]. Furthermore, nuclear speckles are formed through the exchange of constituent RBPs and RNAs with the surrounding nucleoplasm [43]. In addition, the nuclear speckles are reported to be enriched for the SP protein family, which is a set of RBPs named for the IDRs of their serine and arginine residues. For instance, SRRM2, an important RBP in the SR family, was found as a core nuclear speckle scaffold protein, which is required for nuclear speckle formation [68]. In summary, nuclear speckles are one type of self-assembled MLO composed of LLPS-related RBPs or RNAs that mediate multiple critical steps of RNA processing.

Taken together, these findings have highlighted the important roles of LLPS in the formation of MLOs in the nucleus, as well as the LLPS of biomacromolecules participants in the heterochromatin formation and coordination of mRNA processing in the eukaryotic nucleus. Despite LLPS being involved in the physiological formation and maintenance of various MLOs, LLPS triggered by abnormal or mutated proteins is also linked to pathology due to irreversible hydrogelation through amyloid-like aggregation.

## 5. Liquid–Liquid Phase Separation in Metabolic Diseases

A metabolic disorder is a condition that interferes with the normal metabolism process of converting food to energy at the cellular level. Deposition of some aggregated and misfolded amyloids caused by LLPS of aberrant proteins is the hallmark of many metabolic diseases including type 2 diabetes mellitus (T2DM), Alzheimer’s disease (AD), and metabolic bone diseases (MBDs). This section will cover the current understanding of the amyloid formation and LLPS triggered by aberrant proteins during metabolic diseases (Table 2).

### 5.1. Liquid–Liquid Phase Separation and Type 2 Diabetes Mellitus

Type 2 diabetes mellitus (T2DM), accounting for more than 90% of patients with diabetes, is a global chronic metabolic disease. T2DM disease and its complications have resulted in profound psychological and physical distress to patients and brought a huge burden on healthcare systems [78]. Clinically, there are two well-recognized pathogenic factors for T2DM, including insulin resistance and β-cell failure [79]. Notably, emerging evidence showed that the pathologic amyloid deposition by protein misfolding was a hallmark of T2DM. The deposition of misfolded amyloid proteins, concomitantly with the control loss of co-expressed insulin, has been implicated in the failure of pancreatic β-cells in T2DM. Thus, elucidating the amyloidogenesis mechanism is paramount for developing novel therapeutic strategies for T2DM. Increasing evidence demonstrated that the amyloid proteins/peptides could undergo LLPS before the formation of amyloid fibrils through which the amyloids formed irreversible hydrogelation to interfere with normal cellular functions [8]. The LLPS-driven aggregation is a common amyloid feature and integral to pathology. It is reported that dysregulated LLPS of amyloids is closely associated with aberrant protein aggregation in pancreatic β-cells, which leads to T2DM. The islet amyloid polypeptide (IAPP), as the major amyloid component, could undergo LLPS by triggering deleterious aggregation and formation of islet amyloid deposits, which resulted in the pancreatic β-cell dysfunction and was at the heart of the pathology of T2DM [69]. More than 90% of patients with T2DM have misfolded IAPP deposits in the pancreas. Specifically, IAPP is an intrinsically disordered amyloid peptide (IDAP) with the ability to undergo LLPS and misfold to form amyloid fibrils. As a result, the damaged pancreatic β-cell decreased release of insulin, and impaired glucose regulation might be at the heart of the pathology of T2DM [80]. Most importantly, increasing studies pointed out that the inhibition of amyloid aggregation and LLPS has shown initial promise for the therapeutic treatment of T2DM and clinical applications, suggesting IAPP as a potential therapeutic target of T2DM [81]. Additionally, recent research revealed that SG formation contributed to the dysfunction of the β-cell and thus resulted in T2DM [82]. Therefore, it is believed that targeting amyloids and SGs formed by LLPS to block the amyloidosis deposits formation subsequently relieves the β-cell failure, which might be a novel potential therapeutic strategy for T2DM. 

### 5.2. Liquid–Liquid Phase Separation and Alzheimer’s Disease

Alzheimer’s disease (AD), also called type 3 diabetes, is a common metabolic-related neurological disorder without a cure or effective treatments [83]. As a typical protein misfold-inducing disease, AD is characterized by the insoluble intracellular fibrillar structures and the amyloid deposition formed by LLPS, suggesting a strong link between LLPS and the pathogenic process in AD [84] (Figure 3). In fact, several proteins that undergo LLPS and form toxic aggregates have emerged as pathological hallmarks and fundamental mechanisms of AD [85]. TAU, widely reported to play a major role in the development of AD, is a microtubule-associated protein that could undergo LLPS and form intraneuronal neurofibrillary tangles. Specifically, as a type of intrinsically disordered protein (IDP), TAU undergoes LLPS in the dynamic liquid drops and on the surface of microtubules. Thus, TAU could promote microtubule assembly and be hyperphosphorylated in AD. Therefore, the liquid-to-solid transitions of dysfunctional TAU led to the formation of protein tangles and fibrils implicated in AD [70,86]. Notably, it is reported that RNA-binding protein TIA-1 could drive and potentiate TAU LLPS, thus promoting the generation of toxic oligomeric TAU in the brains of AD patients [72], which indicates that TIA-1 might be a potential target for AD. It is acknowledged that amyloid-β (Aβ) plays a key role in AD. The amyloid-β (Aβ) also forms amyloid plaques through LLPS inside cells in the brain and triggers AD pathology [71]. More importantly, Schützmann et al. pointed out that Endo-lysosomal Aβ concentration and low pH are important factors that aggravate α-Syn aggregation and LLPS to assemble as amyloid plaques associated with AD [87]. Furthermore, another RNA-binding protein U1-70K, as an aggregated protein, is mainly concentrated in the brains of AD patients and co-locales with TAU. Xue et al. observed that two fragments in the low-complexity (LC) domain of U1-70K could undergo LLPS forming insoluble depositions in the brains of AD patients. The results firstly identified U1-70K as a new potential target for therapy of AD [73]. More importantly, it appears that multiple MLOs containing different biomacromolecules, including SGs, paraspeckles, and synaptic densities also play a crucial role in the pathology of AD. Numerous studies have shown that the aberrant regulation of SG formation and clearance accelerated the pathological process of AD [88]. Moreover, the assembly of SGs is reported to facilitate the misfolding and propagation of TAU protein, which subsequently caused the pathology of AD [89]. Altogether, the AD condition involves clumps of proteins including tangles of TAU, Aβ plaques, and U1-70K condensates, which undergo LLPS and accumulate in the brains of AD patients. Therefore, understanding the mechanism of LLPS triggered by these proteins is conducive to shedding light on the pathogenesis of AD. Nowadays, the pharmacological intervention of preventing the pathological aggregation of AD-related proteins (TAU, TIA-1, Aβ, U1-70K) is a promising therapeutic strategy for AD.

Emerging evidence has shown that metabolic disturbance seriously affects the induction and progression of neurodegenerative diseases. Metabolic syndrome is becoming an early risk factor for Parkinson’s disease (PD) and even etiology [90,91]. Synuclein (α-Syn) is a natively unstructured protein with the characteristic of LLPS, and its aggregation and amyloid fibrils formation is directly associated with PD pathogenesis [92]. Specifically, α-Syn undergoes LLPS in the presence of a molecular crowder and then forms an amyloid hydrogel that contains oligomers and fibrillar species through a liquid-to-solid transition. It is demonstrated that the N terminus and hydrophobic NAC domain play important roles in driving α-Syn LLPS which is the initial step towards α-Syn aggregation associated with PD pathology. More importantly, low pH, phosphomimetic substitution, and familial PD mutations have been reported to exacerbate α-Syn to undergo LLPS and its aggregation as well as its subsequent amyloid fibrils formation [92].

### 5.3. Liquid–Liquid Phase Separation and Metabolic Bone Diseases

Metabolic bone diseases (MBDs) are the third most common endocrine disorder following diabetes and thyroid diseases. The prevalent MBDs consist mainly of osteoporosis, rickets, and osteomalacia, whereas the rare MBDs comprise Paget’s disease, amyloid bone diseases, osteogenesis imperfecta, and so on [93]. These disorders are commonly caused by abnormal levels of minerals such as calcium or phosphorus, vitamin D, or abnormalities in bone structure. Notably, researchers recently found that pathogenic proteins associated with MBDs could undergo LLPS to form reversible amyloid structures, thus leading to disease pathology.

Osteoporosis is the most common metabolic bone disease characterized by low bone mass and structural deterioration of bone tissue, leading to bone fragility and increased susceptibility to fractures [94]. Accumulating evidence has shown that the patients with AD were 1.79 times more likely to suffer from osteoporosis (AD-associated osteoporosis) than the patients without AD. More importantly, Aβ, the pathological hallmark of AD, is reported to directly suppress the proliferation of bone marrow mesenchymal stem cells (BMSCs) and subsequently inhibit osteogenesis [74]. In fact, Aβ, as an intrinsically disordered protein, could undergo LLPS to aggregate and form the amyloid deposits. Therefore, it might be that the underlying mechanism of Aβ affects the development of AD-associated osteoporosis (Figure 4A). However, another study reported the opposite result that Aβ promoted bone formation by simultaneously enhancing osteogenic differentiation of BMSCs and inhibiting osteoclast differentiation [95]. It is possible that the roles of Aβ vary in different osteoporosis, which might be related to differences in osteoporosis etiology.

Recently, Liu et al. reported that the mechanosensitive lncRNA *Neat1* promoted osteoblast function through MLO paraspeckle driven by LLPS [75]. Moreover, the sealing zone, an osteoclast-specific cytoskeletal structure, is a specialized cell-matrix adhesion structure as a type of MLO. The structure is critical for osteoclast-mediated bone resorption by demarcating the area of bone resorption from the rest of the environment [96]. The findings indicated the significant roles of LLPS and MLOs in bone remodeling and osteoporosis, which might provide a novel therapeutic strategy for osteoporosis and several possible targets for new drugs. Additionally, Paget’s disease of bone (PDB) is a progressive condition characterized by haphazard and improper bone turnover, leading to larger and weaker bones than normal. PDB affects ~3.1% of individuals older than 55 years in the United Kingdom [97]. Nevertheless, the cause and mechanism of PDB are still unknown and controversial. Recent studies reported that the *p62* gene, a common component and early marker of protein aggregates, could undergo LLPS to form the spherical liquid-like droplets, namely the p62 body, thereby controlling cellular physiological processes [98]. Notably, several studies have identified that the germline mutations of the *p62* gene in PDB patients compromised LLPS and p62 body formation [76,99]. Altogether, LLPS triggered by dysfunctional or mutational *p62* gene might in part lead to PDB (Figure 4B).

Another important amyloid β2-Microglobulin (β2M) plays an important role in dialysis-related amyloid bone diseases. Amyloid bone diseases are the major complication of chronic renal failure and long-term renal replacement therapy [100]. Several pieces of evidence suggested that the amyloid fibrils of β2M deposit in bones and joints were mainly present in carpal tunnel syndrome, destructive arthropathy, subchondral bone erosions, and osteoarthritis [77]. Although there is no evidence that β2M could undergo LLPS, we speculated that β2M with the characteristics of aggregation might form the pathologic amyloid deposition through LLPS. Moreover, the latest study revealed that core regulatory circuitry (CRC) components undergo LLPS to form condensates. Thus, it might be an important mechanism of CRC-mediated metastasis and chemoresistance in osteosarcoma [101]. They also found that the pharmacological inhibition of LLPS of CRC components could effectively intervene in the chemo-resistant and metastatic osteosarcoma, which hinted that the LLPS-based pharmacological strategy is a novel potential therapeutic method for human diseases.

Taken together, the multiple proteins with the characteristics of LLPS could assemble into the condensates or amyloid fibrils and are closely associated with the pathology of various BMDs. Importantly, these findings suggested the significant roles of LLPS in novel molecular mechanisms and treatment strategies for metabolic bone diseases. Moreover, it would be interesting to examine the deep interaction between LLPS and metabolic bone diseases and thus explore the novel pathogenic mechanisms of metabolic bone diseases.

## 6. Concluding Remarks and Future Perspectives

LLPS is emerging as a key mechanism that mediates the biogenesis of functional MLOs and the formation of aggregated structures linked to several human diseases. It is now established that several pathological proteins are also able to undergo LLPS and are aggregated in the insoluble deposition in metabolic diseases. Likewise, metabolic disease-related mutations and conditions also alter the LLPS behavior of these proteins, which could elicit toxicity. Therefore, the pharmacological interventions that antagonize the LLPS of aberrant proteins might be able to mitigate toxicity and aggregation in metabolic diseases.

Here, we have reviewed the development and roles in MLO formation by LLPS and discussed the mechanisms of the LLPS of pathologic proteins in contributing to metabolic diseases, especially type 2 diabetes mellitus, Alzheimer’s disease, and metabolic bone diseases. Despite some similarities between these diseases, the pathologic proteins obey a distinct mode of molecular regulation and aggregate in different forms of LLPS. The function of LLPS in future studies is likely to continue to uncover new mechanisms and therapeutic strategies for these diseases.

Although the LLPS field is growing rapidly and is still in the exploratory stage, the mechanism underlying LLPS has indeed updated our understanding of biological activities and disease conditions. Further studies on the roles of LLPS in metabolic diseases are expected to unravel the novel pathology of metabolic diseases and improve their clinical therapeutics.

## Figures and Tables

**Figure 1 cells-11-03023-f001:**
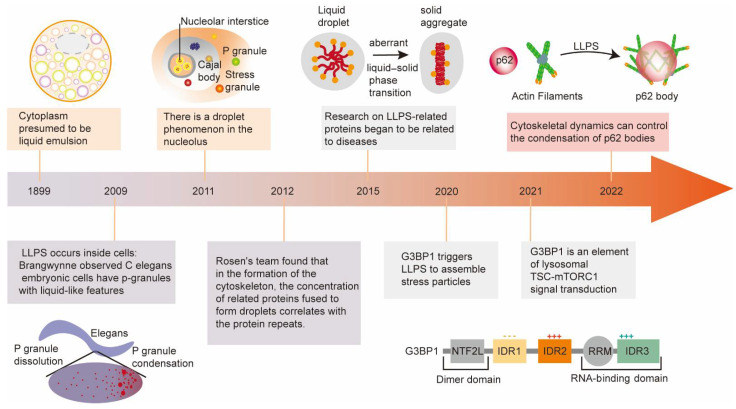
The development history and discovery of the amazing and vital roles of LLPS in biology. Representative milestones sparking tremendous development of LLPS are enumerated in the figure.

**Figure 2 cells-11-03023-f002:**
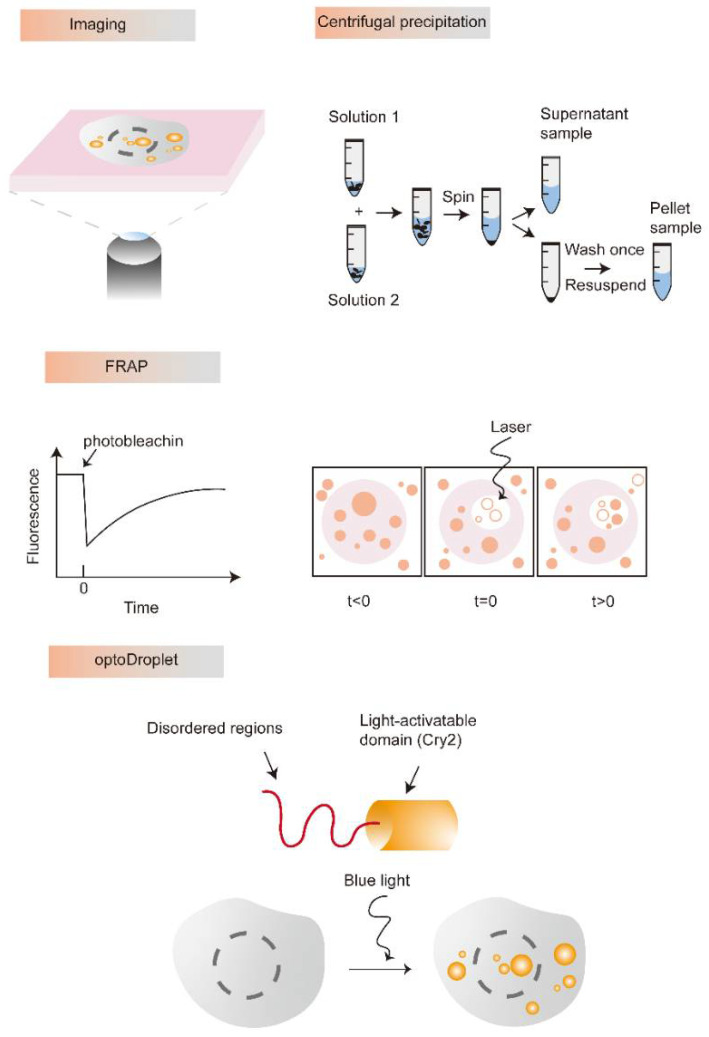
Representative research methods and technology to identify or study LLPS. Various microscopic techniques can be used to detect the process of phase transition and visualize the properties of droplets. Centrifugal precipitation is another common detection strategy of LLPS. The FRAP is the well-recognized method for the observation of LLPS, which was accomplished by measuring the fluorescence intensity of the bleached region prior to, immediately after, and throughout recovery from bleaching. The optoDroplet provides a level of control that we can use to precisely map the phase diagram in living cells.

**Figure 3 cells-11-03023-f003:**
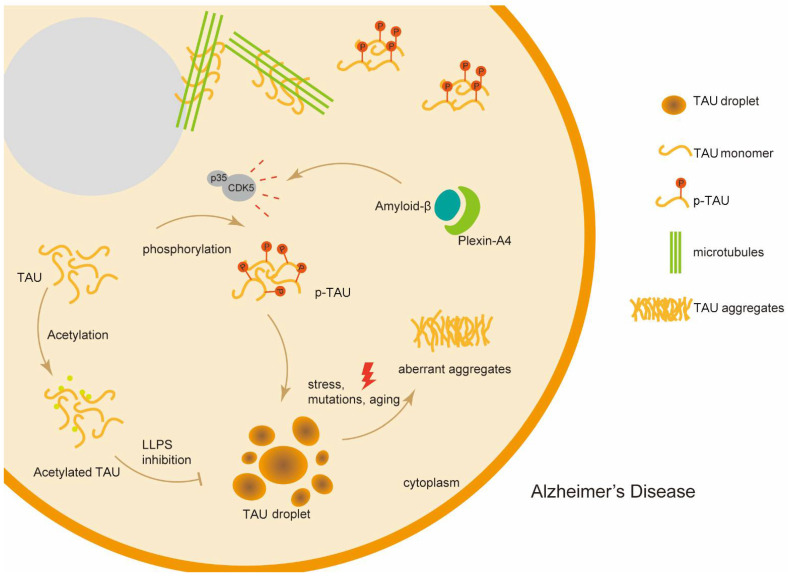
Amyloid proteins TAU and Aβ aggregates in brains of Alzheimer’s disease patients (AD). Post-transcriptional control, stress, or mutations altered the behaviors of TAU-triggered LLPS, which subsequently induced aberrant aggregates of TAU and resulted in pathology of AD. Aβ aggregated as amyloid plaques through LLPS and interacted with plexin-A4 proteins thus leading to AD.

**Figure 4 cells-11-03023-f004:**
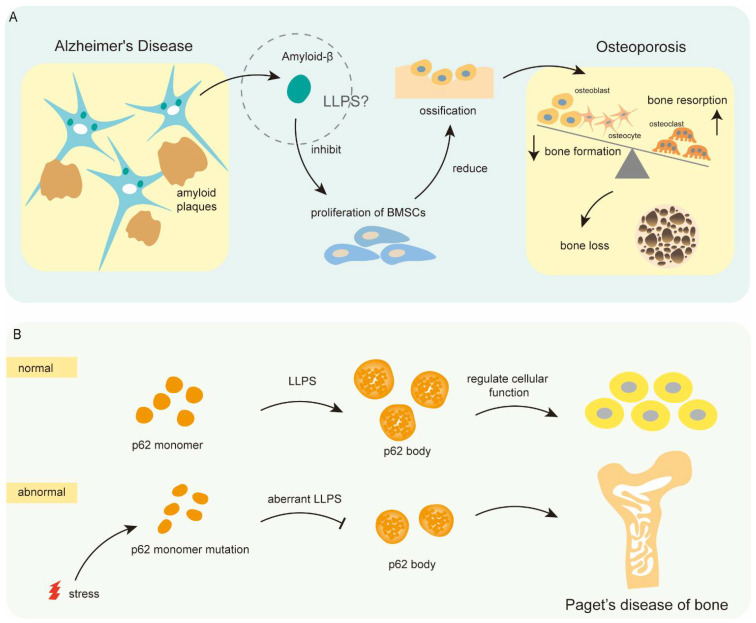
Roles of Amyloid proteins Aβ and p62 in metabolic bone diseases. (**A**) Aberrant aggregation of Amyloid-β through LLPS in the brain of AD patients can result in decreased ossification by inhibiting the proliferation of BMSCs. (**B**) Mutation of p62 led to its aberrant LLPS which further affected the formation of p62 body and resulted in pathology of Paget’s disease of bone.

**Table 1 cells-11-03023-t001:** Examples of the various MLOs formed by LLPS and their functions.

Localization	Names of Condensates	Biological Function	References
Plasma membrane	TCR clusters	T-cell immune signal transduction	[32]
	Nephrin clusters	Glomerular filtration barrier	[11]
	Actin patches	Endocytosis	[33]
	Focal adhesions	Cell adhesion and migration	[16]
	Synaptic densities	Neurotransmission	[26]
Cytoplasm	Stress granules	mRNA storage and translational regulation	[34]
	RNA transport granules	mRNA storage and transport in neuronal cells	[35]
	U body	Storage and assembly of snRNPs	[36]
	P body	mRNA decay and silencing	[37]
	Balbiani body	A transient collection of proteins, RNA, and membrane-bound organelles found in primary oocytes of all animals observed to date	[38]
	P granules	Germ cell lineage maintenance in Caenorhabditis elegans	[10]
Nucleus	cGAS condensates	Innate immune signaling	[39]
	Cleavage body	mRNA processing	[40]
	Cajal body	Assembling spliceosomal small nuclear ribonucleoproteins	[40,41]
	Nucleoli	rRNA storage, rRNA synthesis and processing, and assembly of ribosomal subunits	[41]
	Gem	Aid histone mRNA processing	[42]
	Nuclear speckles	mRNA splicing	[43]
	OPT domain	Transcriptional regulation	[44]
	PcG body	Transcriptional repression	[45]
	PML bodies	Apoptotic signaling, anti-viral defense, and transcription regulation	[46]
	Histone locus body	Processing of histone mRNAs	[47]
	Paraspeckles	Storage of certain RNAs	[48]
	Perinucleolar compartment	Related to malignancy	[49]

**Table 2 cells-11-03023-t002:** Roles of LLPS in metabolic diseases.

Type of Disease	Connection with LLPS	Substances Involved	References
Type 2 Diabetes Mellitus (T2DM)	Islet amyloid polypeptide (IAPP) undergoes AWI-catalyzed LLPS, which initiates hydrogelation and aggregation	Islet amyloid polypeptide (IAPP)	[69]
Alzheimer’s disease (AD)	Aberrant deposition of TAU protein in the brain triggered by LLPS	RNA binding protein TAU	[70]
	Amyloid-β (Aβ) forms amyloid plaques through LLPS inside cells in the brain	Amyloid-β (Aβ)	[71]
	TIA-1 potentiates TAU LLPS thus promoting the generation of toxic oligomeric TAU in brains	TIA-1	[72]
	U1-70K undergoes LLPS forming insoluble depositions in brains	RNA binding protein U1-70K	[73]
Metabolic bone diseases (MBDs)	Aβ undergoes LLPS and form amyloid deposits to induce AD-associated osteoporosis	Amyloid-β (Aβ)	[74]
	LncRNA *Neat1* promotes osteoblast function through MLO paraspeckle	LncRNA *Neat1*	[75]
	Germline mutations of the P62 gene in PDB patients that compromise LLPS and P62 body formation	*P62*	[76]
	Amyloid fibrils of β2M deposit in bones and joints	β2-Microglobulin (β2M)	[77]

## Data Availability

Not applicable.

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
