# Peer review of "Liquid–Liquid Phase Separation of Biomacromolecules and Its Roles in Metabolic Diseases"

_cells, 2022, doi:10.3390/cells11193023_

Round 1
Reviewer 1 Report
LLPS is an expanding field that has been reviewed several times recently [1-3]. The idea that LLPS contributes to metabolic diseases is intriguing and important. However, the authors dilute this concept with a lengthy exposition of the history of LLPS, stress granules, P bodies, nucleoli and speckles that can be found in many other reviews. This part could be summarized in two short paragraphs. On the other hand, synuclein aggregation could have been discussed PMID: 32514159. Another omission in light of the role of amyloids in all diseases discussed is the transition from soluble condensates to fibrils and the rules that control that transition.
In addition, there are several writing mistakes. For example:
In the abstract this sentence is not clear: Aberrant LLPS caused by stress or pathological conditions can disrupt the metabolic systems by distinct machinery.
Lane 35: P bodies
Lanes 46-47 Nevertheless, it is currently unclear that what
Lane 52: increasing experiments: this is not clear; recent experiments ? this is repeated in other parts
Lane 123: Researchers religiously believe: this is not scientific writing
Lane 410: is specifically aggregates
1. Nesterov, S.V.; Ilyinsky, N.S.; Uversky, V.N. Liquid-liquid phase separation as a common organizing principle of intracellular space and biomembranes providing dynamic adaptive responses. Biochim Biophys Acta Mol Cell Res 2021, 1868, 119102, doi:10.1016/j.bbamcr.2021.119102.
2. Su, Q.; Mehta, S.; Zhang, J. Liquid-liquid phase separation: Orchestrating cell signaling through time and space. Mol Cell 2021, 81, 4137-4146, doi:10.1016/j.molcel.2021.09.010.
3. Alberti, S.; Hyman, A.A. Biomolecular condensates at the nexus of cellular stress, protein aggregation disease and ageing. Nat Rev Mol Cell Biol 2021, 22, 196-213, doi:10.1038/s41580-020-00326-6.
Author Response
Thank you for your precious comments and advice. Those comments are all valuable and very helpful for revising and improving our paper, as well as the important guiding significance to our researches. We have studied comments carefully and have made correction which we hope meet with approval. The main corrections in the paper and the responds to the reviewer’s comments are as following:
LLPS is an expanding field that has been reviewed several times recently [1-3]. The idea that LLPS contributes to metabolic diseases is intriguing and important.
Response:Thank you for your summary. We really appreciate your efforts in reviewing our manuscript. We have revised the manuscript accordingly. Our point-by-point responses are detailed below.
However, the authors dilute this concept with a lengthy exposition of the history of LLPS, stress granules, P bodies, nucleoli and speckles that can be found in many other reviews. This part could be summarized in two short paragraphs.On the other hand, synuclein aggregation could have been discussed PMID: 32514159. Another omission in light of the role of amyloids in all diseases discussed is the transition from soluble condensates to fibrils and the rules that control that transition.
Response:Thank you for your summary. We really appreciate your efforts in reviewing our manuscript. We have revised the manuscript accordingly. We have summarized the history of LLPS and the section of membrane-less organelles (MLOs) in short paragraphs. Moreover, we discussed the synuclein aggregation and the transition from soluble condensates to fibrils and the rules that control that transition, which can be found in the revised manuscript.
In addition, there are several writing mistakes. For example:
In the abstract this sentence is not clear: Aberrant LLPS caused by stress or pathological conditions can disrupt the metabolic systems by distinct machinery.
Response:We are extremely grateful to you for pointing out this problem. We apologize for our error and we have changed the descriptions. We have deleted this sentence and reedited the abstract in the revised manuscript.
Lane 35: P bodies
Response:We are extremely grateful to you for pointing out this problem. We apologize for our error and we have changed the descriptions. P bodies has been corrected as Processing bodies (P-bodies).
Lanes 46-47 Nevertheless, it is currently unclear that what
Response:Thank you for your careful review. This sentence has been replaced with “However, the in-depth mechanism of metabolic diseases is not yet completely understood.”
Lane 52: increasing experiments: this is not clear; recent experiments? this is repeated in other parts
Response:We are grateful for your suggestion. These words are mis-expressed. We have replaced them with “emerging evidence”.
Lane 123: Researchers religiously believe: this is not scientific writing
Response:Thank you for underlining this deficiency. This sentence is not scientific writing and we have replaced it with “Some proponents even said that LLPS might rewrite the rules of drug development.”
Lane 410: is specifically aggregates
Response:We are extremely grateful to you for pointing out this problem. We apologize for our error and we have changed the descriptions. “is specifically aggregates” has been replaced with “is mainly concentrated in brain”
Thank you for your careful review. We really appreciate your efforts in reviewing our manuscript during this unprecedented and challenging time. We wish good health to you, your family, and community. Your careful review has helped to make our study clearer and more comprehensive.
Reviewer 2 Report
I read the review by Chen et al. with interest. The phenomenon of liquid-liquid phase separation of biomolecules in cells, rediscovered in 2009, underlies the formation of membraneless organelles (MLOs). MLOs are highly dynamic and at the same time highly organized structures that are easily assembled and disassembled under the influence of various external and internal stimuli in the cell, and which have proven to be extremely important for the life and functioning of cells. Since 2009, the study of the formation and functions of MLOs in normal and in various diseases has been a top trend, and according to PudMed data, the number of articles has grown exponentially over the past 3-5 years. Therefore, this review is undoubtedly relevant. The authors reviewed the current literature on the role of the occurrence of certain defects in a number of MLOs that lead to the emergence of amyloid fibrils associated with metabolic diseases, thereby revealing the cause of these diseases, and, consequently, identifying completely new approaches for the treatment of diseases associated with the formation of amyloid fibrils. It seemed to me especially interesting that the authors consider Alzheimer's disease as a metabolic disease. Usually in the literature this disease is referred to as a neurodegenerative disease and is associated with amyloid plaques. A new look at this disease can bring closer understanding of the causes of this disease, and, consequently, the ways of its prevention and treatment. The work is well written and presented. I have only small comments:
Minor comments:
1) Liquid-Liquid Phase Separation is a physical phenomenon. It cannot be good or bad, right or wrong. Line 112 and 117 "abnormal LLPS". However, I have seen "aberrant LLPS" in the literature.
In my opinion it is important what exactly is exposed to LLPS, e.g. normal or mutant proteins. It may be also that due to LLPS a functional membraneless organelle (MLO) is assembled, but then it passes into a gel state, and then into amyloid fibrils. Thus, it seems to me that the essence of the problem lies in MLO.
In general, I suggest the authors to think about this again.
Writing "...LLPS has made many brilliant achievements in the field of biology.." line 67 is definitely bad.
2) In Section 2. many of the statements are written naively carelessly or simply badly (e.g. lines 63-65, 84-87, 89-91)
3) Section 3. There are many more approaches to examine MLO. (see, e.g. Antifeeva et al., Cellular and Molecular Life Sciences (2022) 79:251). You can limit yourself to describing those that are given, but in this case it must be mentioned that these are the most common methods.
4) Figure 2, actually contains three panels that must be indicated: FRAP method is described in the text and in the caption, observation using microscopy (imaging) and precipitation are mentioned in the text, but not in the caption. In the middle panel the formation of optoDroplets induced by light is given. Nothing is said about this either in the text or in the capture. It is also not clear why this is given among the methods for studying MLOs.
5) Section 4. Dozens of MLOs are known for to date. The title of the table correctly states that these are examples of MLOs. It should also be emphasized in the text that only examples of MLOs with different localizations are given.
6) I propose to rephrase the sentence: “It is well recognized that LLPS of biomacromolecules seems to have emerged as a ubiquitous route to form MLOs in living cells” (Line 187).
7) Line 379 “.... an intrinsically disordered amyloid peptides (IDPs)”. Firstly the abbreviation must be (IDAPs), in addition IDPs is a well-established abbreviation for intrinsically disordered proteins.
8) Ref. 88 does not consider LLPS. The statement in Table 2 and lines 408-409 is incorrect.
Author Response
Thank you for your precious comments and advice. Those comments are all valuable and very helpful for revising and improving our paper, as well as the important guiding significance to our researches. We have studied comments carefully and have made correction which we hope meet with approval. The main corrections in the paper and the responds to the reviewer’s comments are as following:
I read the review by Chen et al. with interest. The phenomenon of liquid-liquid phase separation of biomolecules in cells, rediscovered in 2009, underlies the formation of membraneless organelles (MLOs). MLOs are highly dynamic and at the same time highly organized structures that are easily assembled and disassembled under the influence of various external and internal stimuli in the cell, and which have proven to be extremely important for the life and functioning of cells. Since 2009, the study of the formation and functions of MLOs in normal and in various diseases has been a top trend, and according to PudMed data, the number of articles has grown exponentially over the past 3-5 years. Therefore, this review is undoubtedly relevant. The authors reviewed the current literature on the role of the occurrence of certain defects in a number of MLOs that lead to the emergence of amyloid fibrils associated with metabolic diseases, thereby revealing the cause of these diseases, and, consequently, identifying completely new approaches for the treatment of diseases associated with the formation of amyloid fibrils. It seemed to me especially interesting that the authors consider Alzheimer's disease as a metabolic disease. Usually in the literature this disease is referred to as a neurodegenerative disease and is associated with amyloid plaques. A new look at this disease can bring closer understanding of the causes of this disease, and, consequently, the ways of its prevention and treatment. The work is well written and presented.
Response:We are very grateful to your comments for the manuscript. We appreciate your positive evaluation of our work. According to your advice, we amended the relevant part in manuscript. All of your questions were answered one by one.
Minor comments:
1) Liquid-Liquid Phase Separation is a physical phenomenon. It cannot be good or bad, right or wrong. Line 112 and 117 "abnormal LLPS". However, I have seen "aberrant LLPS" in the literature. In my opinion it is important what exactly is exposed to LLPS, e.g. normal or mutant proteins. It may be also that due to LLPS a functional membraneless organelle (MLO) is assembled, but then it passes into a gel state, and then into amyloid fibrils. Thus, it seems to me that the essence of the problem lies in MLO. In general, I suggest the authors to think about this again. Writing "...LLPS has made many brilliant achievements in the field of biology.." line 67 is definitely bad.
Response:Thank you for underlining this deficiency. As you said, Liquid-Liquid Phase Separation is a physical phenomenon, it cannot be good or bad, right or wrong. The biological roles of this phenomenon are mainly depended on the normal or mutant proteins which drive LLPS. Therefore, we have revised the inappropriate expression in the manuscript and replaced with “LLPS of aberrant proteins”.
The sentence in line 67 “...LLPS has made many brilliant achievements in the field of biology.” has been revised and replaced with sentence “Although increasing research on LLPS has identified its essential roles in physiology and diseases,”
2) In Section 2. many of the statements are written naively carelessly or simply badly (e.g. lines 63-65, 84-87, 89-91)
Response:Our deepest gratitude goes to you for your careful work and thoughtful suggestions that have helped improve this paper substantially. We apologize for the language problems in the original manuscript. The language presentation was improved with assistance from a native English speaker.
3) Section 3. There are many more approaches to examine MLO. (see, e.g. Antifeeva et al., Cellular and Molecular Life Sciences (2022) 79:251). You can limit yourself to describing those that are given, but in this case it must be mentioned that these are the most common methods.
Response:We deeply appreciate your suggestion. We have carefully read the literature and some other common methods have been also reviewed in the revised manuscript (Ref 28).
4) Figure 2, actually contains three panels that must be indicated: FRAP method is described in the text and in the caption, observation using microscopy (imaging) and precipitation are mentioned in the text, but not in the caption. In the middle panel the formation of optoDroplets induced by light is given. Nothing is said about this either in the text or in the capture. It is also not clear why this is given among the methods for studying MLOs.
Response:Thank you for your careful review. We are very sorry for the mistakes in this manuscript and inconvenience they caused in your reading. We have carefully revised the second part of the manuscript and the Figure 2 is recomposed.
5) Section 4. Dozens of MLOs are known for to date. The title of the table correctly states that these are examples of MLOs. It should also be emphasized in the text that only examples of MLOs with different localizations are given.
Response:We are extremely grateful to you for pointing out this problem. We have emphasized in the manuscript that only examples of MLOs with different localizations are given in the present review.
6) I propose to rephrase the sentence: “It is well recognized that LLPS of biomacromolecules seems to have emerged as a ubiquitous route to form MLOs in living cells” (Line 187).
Response:Thank you for underlining this deficiency. We apologize for not describing the sentence clearer. We have rephrased the sentence. We have rephrased the above sentence as “It is well recognized that LLPS of biomacromolecules have emerged as a biophysical basis for the formation of MLOs in living cells”
7) Line 379 “.... an intrinsically disordered amyloid peptides (IDPs)”. Firstly the abbreviation must be (IDAPs), in addition IDPs is a well-established abbreviation for intrinsically disordered proteins.
Response:Our deepest gratitude goes to you for your careful work and thoughtful suggestions that have helped improve this paper substantially. We have carefully checked the abbreviation of “intrinsically disordered amyloid peptides, IDAPs” and “intrinsically disordered proteins, IDPs”.
8) Ref. 88 does not consider LLPS. The statement in Table 2 and lines 408-409 is incorrect.
Response:We are extremely grateful to you for pointing out this problem. We are very sorry for the description and inconvenience they caused in your reading. We have carefully checked the Reference 88 (Liquid–liquid phase separation of type II diabetes-associated IAPP initiates hydrogelation and aggregation) and found that they have demonstrated that IAPP could undergo AWI-catalyzed LLPS, which triggers T2DM pathological aggregation.
Thank you for your precious comments and advice. Our deepest gratitude goes to you for your careful work and thoughtful suggestions that have helped improve this paper substantially. We wish good health to you, your family, and community.
Reviewer 3 Report
The review article by Yang and co-workers summarizes current knowledge on the principles of liquid-liquid phase separation and its role in a group of diseases.
The organization of the review is logical and clear. The illustrations nicely recapitulate the key concepts.
The one limitation is that the overview of the different systems comes across as a bit rote, in that there is little critical examination of the discussed works. Therefore the manuscript appears rather dry and at times not too informative. One suggestion for improvement would be to better point out, on the one hand, those conclusions that are supported by extensive evidence, and on the other hand the findings that still warrant confirmation. For example: is it definitively demonstrated that LLPS is the (only) cause of aggregation of tau? Condensates are simply concomitant or causative of protein transformations?
The authors focus on metabolic disorders, however there is little reference to metabolic dysfunction, instead the main theme concerns the aggregation of proteins. I suggest expounding on the (possible) connection between protein aggregation and metabolism.
Minor point: In Figure 2 several experimental techniques are presented, however the caption describes only FRAP. Please complete the caption.
The text is well understandable, however it is not fluent and the language should be improved.
Author Response
Thank you for your precious comments and advice. Those comments are all valuable and very helpful for revising and improving our paper, as well as the important guiding significance to our researches. We have studied comments carefully and have made correction which we hope meet with approval. The main corrections in the paper and the responds to the reviewer’s comments are as following:
The review article by Yang and co-workers summarizes current knowledge on the principles of liquid-liquid phase separation and its role in a group of diseases. The organization of the review is logical and clear. The illustrations nicely recapitulate the key concepts.
Response: We are very grateful to your comments for the manuscript. We appreciate your positive evaluation of our work. According to your advice, we amended the relevant part in manuscript. All of your questions were answered one by one.
The one limitation is that the overview of the different systems comes across as a bit rote, in that there is little critical examination of the discussed works. Therefore the manuscript appears rather dry and at times not too informative. One suggestion for improvement would be to better point out, on the one hand, those conclusions that are supported by extensive evidence, and on the other hand the findings that still warrant confirmation. For example: is it definitively demonstrated that LLPS is the (only) cause of aggregation of tau? Condensates are simply concomitant or causative of protein transformations?
Response: Thank you for underlining this deficiency. We apologize for not describing the manuscript clearer. We have summarized the history of LLPS and the section of membrane-less organelles (MLOs) in short paragraphs. The conclusions that are supported by extensive evidence have added in the revised manuscript. Thank you very much.
The authors focus on metabolic disorders, however there is little reference to metabolic dysfunction, instead the main theme concerns the aggregation of proteins. I suggest expounding on the (possible) connection between protein aggregation and metabolism.
Response:We deeply appreciate your suggestion. According to the reviewer’s comment, we have summarized the history of LLPS, stress granules, P bodies, nucleoli and speckles in short paragraphs. Moreover, we expounded on the (possible) connection between protein aggregation and metabolism in the revised manuscript.
Minor point: In Figure 2 several experimental techniques are presented, however the caption describes only FRAP. Please complete the caption.
Response:Thank you for your careful review. We have carefully revised the second part of the manuscript and the Figure 2 is recomposed.
The text is well understandable, however it is not fluent and the language should be improved.
Response:Thank you for your careful review. We are very sorry for the mistakes in this manuscript and inconvenience they caused in your reading. The manuscript has been thoroughly revised and rewritten by a native English speaker, so we hope it could meet the journal’s standard.
We really appreciate your efforts in reviewing our manuscript during this unprecedented and challenging time. We wish good health to you, your family, and community. Your careful review has helped to make our study clearer and more comprehensive.
Round 2
Reviewer 1 Report
I am satisfied with the authors' response
Reviewer 3 Report
The authors have addressed my concerns.
I find the language still needs some correction.